

# Impact of Front Range sources on reactive nitrogen concentrations and deposition in Rocky Mountain National Park

Katherine B. Benedict[1], Anthony J. Prenni[2], Amy P. Sullivan[1],
Ashley R. Evanoski-Cole[1,3], Emily V. Fischer[1], Sara Callahan[1], Barkley C. Sive[2],
Yong Zhou[1], Bret A. Schichtel[4] and Jeffrey L. Collett Jr[1]

[1] Department of Atmospheric Science, Colorado State University, Fort Collins, CO, United States of America
[2] Air Resources Division, National Park Service, Lakewood, CO, United States of America
[3] Present address: Department of Chemistry, St. Bonaventure University, St. Bonaventure, NY, United States of America
[4] Air Resources Division, National Park Service, Fort Collins, CO, United States of America

Corresponding author
Katherine B. Benedict,
katherine.benedict@colostate.edu

## ABSTRACT

Human influenced atmospheric reactive nitrogen (RN) is impacting ecosystems in Rocky Mountain National Park (ROMO). Due to ROMO's protected status as a Class 1 area, these changes are concerning, and improving our understanding of the contributions of different types of RN and their sources is important for reducing impacts in ROMO. In July–August 2014 the most comprehensive measurements (to date) of RN were made in ROMO during the Front Range Air Pollution and Photochemistry Éxperiment (FRAPPÉ). Measurements included peroxyacetyl nitrate (PAN), $C_1$–$C_5$ alkyl nitrates, and high-time resolution $NO_x$, $NO_y$, and ammonia. A limited set of measurements was extended through October. Co-located measurements of a suite of volatile organic compounds provide information on source types impacting ROMO. Specifically, we use ethane as a tracer of oil and gas operations and tetrachloroethylene ($C_2Cl_4$) as an urban tracer to investigate their relationship with RN species and transport patterns. Results of this analysis suggest elevated RN concentrations are associated with emissions from oil and gas operations, which are frequently co-located with agricultural production and livestock feeding areas in the region, and from urban areas. There also are periods where RN at ROMO is impacted by long-range transport. We present an atmospheric RN budget and a nitrogen deposition budget with dry and wet components. Total deposition for the period (7/1–9/30) was estimated at 1.58 kg N/ha, with 87% from wet deposition during this period of above average precipitation. Ammonium wet deposition was the dominant contributor to total nitrogen deposition followed by nitrate wet deposition and total dry deposition. Ammonia was estimated to be the largest contributor to dry deposition followed by nitric acid and PAN (other species included alkyl nitrates, ammonium and nitrate). All three species are challenging to measure routinely, especially at high time resolution.

## INTRODUCTION

The nitrogen cycle has been altered by increasing production of reactive nitrogen (RN) for food production and energy (*Galloway et al., 2008*). Much of the RN produced ultimately ends up in air and water, potentially contributing to environmental problems (*Fenn et al., 1998*; *Howarth, 2004*). Direct leaching from fertilized soils contributes to eutrophication of fresh water. Emissions to the atmosphere can increase ozone and particle formation as well as contribute to elevated nitrogen deposition to land surfaces and water bodies. Elevated nitrogen deposition can alter ecosystems and the services they provide in many regions (*Goulding et al., 1998*; *Fenn et al., 2003*; *Burns, 2004*; *Bell et al., 2017*) as many ecosystems are nitrogen limited (*LeBauer & Treseder, 2008*). Across the globe there has been intense focus on understanding levels of nitrogen deposition and its impacts in particularly sensitive and protected ecosystems. Over the past several decades, researchers in Europe, the United States, and China have focused on better understanding the interactions and changes in ecosystems that occur as a result of excess RN deposition using long-term networks. An extensive network was established throughout Europe in 2006–2011 to measure ambient RN concentrations, estimate dry deposition and provide a comprehensive data set for regional models (*Owen et al., 2011*; *Flechard et al., 2011*). In contrast, work in the United States has focused primarily on wet nitrogen deposition measurements made by the National Atmospheric Deposition Program (NADP) (*Lehmann, Bowersox & Larson, 2005*). In China there has not been a long-term national deposition monitoring program; however there has been an effort to implement long-term standardized nitrogen deposition measurements (*Liu et al., 2011*).

In the United States, one area of focus has been aimed at Rocky Mountain National Park (ROMO), a high elevation site which includes alpine tundra and lakes, exposed bedrock, and subalpine forests (*Baron et al., 1994*). The Rocky Mountains are more susceptible to RN deposition than many other areas because nitrogen is not well retained in the local soil and plants. Reduced plant nitrogen demand results from a short growing season, which decreases nitrogen retention. Additionally, the soil types in the area (exposed bedrock, talus, coarse soils) reduce contact time between drainage water and soil (*Fenn et al., 1998*).

Research aimed at improved understanding of nitrogen deposition in ROMO has been ongoing for many years (*Bowman & Steltzer, 1998*; *Burns, 2003*; *Burns, 2004*; *Baron et al., 2004*; *Baron, 2006*; *Beem et al., 2010*; *Benedict et al., 2013c*; *Benedict et al., 2013a*). Several studies have demonstrated that increased nitrogen deposition has been associated with changes in biological species and ecological processes (*Bowman & Steltzer, 1998*; *Burns, 2004*; *Baron, 2006*), and a high spatial variability of nitrogen deposition in the region has been observed from national monitoring network data (*Burns, 2003*; *Beem et al., 2010*). Measurements characterizing many RN species are routinely collected in the park as part of NADP, the Interagency Monitoring of Protected Visual Environments (IMPROVE) program, the Clean Air Status and Trends Network (CASTNET), and the Ammonia Monitoring Network (AMoN). These networks provide the data needed for tracking progress toward reduction of nitrogen deposition in ROMO, a goal mutually set by the State of Colorado, the US EPA, and the National Park Service (*CDPHE, 2017*).

Special studies like the Rocky Mountain Atmospheric Nitrogen and Sulfur (RoMANS) Studies (*Malm et al., 2009*) have increased the understanding of the type and frequency of atmospheric transport that plays a role in atmospheric nitrogen inputs into the ecosystems at ROMO, as well as the importance of wet organic nitrogen deposition and dry deposition of ammonia ($NH_3$) (*Beem et al., 2010*; *Benedict et al., 2013c*; *Benedict et al., 2013a*).

The Front Range Air Pollution and Photochemistry Experiment (FRAPPÉ) was conducted July 15 through August 18, 2014. Although the primary goal of FRAPPÉ was to better characterize summertime air quality in the Northern Front Range Metropolitan Area (NFRMA), the extensive suite of measurements collected during FRAPPÉ allowed for determination of the degree to which pollution from NFRMA sources and long-range transport contribute to air quality impacts at ROMO. FRAPPÉ was conducted in parallel with the 2014 DISCOVER-AQ (Deriving Information on Surface Conditions from Column and Vertically Resolved Observations Relevant to Air Quality) project in Colorado. Research flights, surface mobile measurements, and stationary ground sites were coordinated to provide a comprehensive picture of the atmospheric processes impacting air quality in the northern Front Range. The majority of operations took place along the Front Range, near the main pollutant sources in the region, although research flights and several ground sites characterized more remote areas to better understand the transport and impact from the source regions.

In this manuscript, we describe measurements made during FRAPPÉ from a ground site collocated with routine monitoring measurements at the Rocky Mountain National Park Longs Peak (ROMO-LP) field site. Measurements began July 7, prior to the FRAPPÉ study period, and continued through October 15, well after the campaign ended. These measurements included high time resolution (1 min–1 h) $NH_3$, nitric oxide (NO), and $NO_y$ ($\sum$oxidized nitrogen) concentrations, as well as more comprehensive 24-h gas ($NH_3$, nitric acid—$HNO_3$, sulfur dioxide—$SO_2$) and particle chemistry (ammonium—$NH_4^+$, nitrate—$NO_3^-$, sulfate—$SO_4^{2-}$, sodium—$Na^+$, potassium—$K^+$, magnesium—$Mg^{2+}$, calcium—$Ca^{2+}$, and chloride—$Cl^-$) measurements. Two gas chromatographs were used to measure select organic nitrogen species including peroxyacetyl nitrate (PAN), methyl nitrate ($MeONO_2$), ethyl nitrate ($EtONO_2$), i-propyl nitrate ($iPrONO_2$), 2-butyl nitrate ($2BuONO_2$), 2-pentyl nitrate ($2PenONO_2$), and 3-pentyl nitrate ($3PenONO_2$); as well as volatile organic compound (VOC) tracers for a variety of sources. These measurements were aimed at better understanding the sources that contribute to pollutant concentrations in ROMO, specifically for RN species. We provide a comprehensive budget for RN in ROMO during FRAPPÉ, calculate a nitrogen deposition budget, and assess important sources for the primary components of the RN budget.

## METHODS

The measurement site at ROMO (Fig. 1) was located south of Estes Park, CO, near the base of Longs Peak. The site ($40.2783°$, $−105.5457°$; 2,784 m ASL) is co-located with IMPROVE (ROMO1) and CASTNet (ROM206/406) national network monitoring sites, and the National Park Service Gaseous Pollutant Monitoring Network (GPMP). Measurements

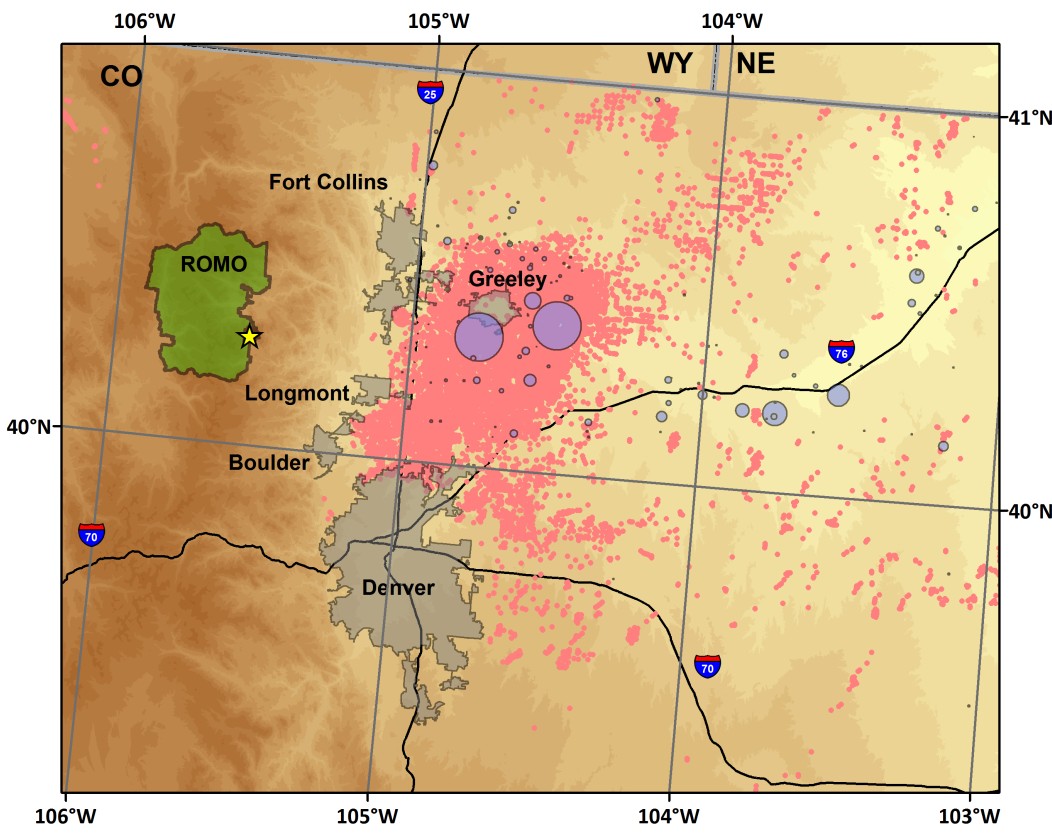

**Figure 1** **Map of the study region.** Included are the sampling site (star), cities along the northern Front Range of Colorado, oil and gas wells (pink) (*COGCC, 2018*), and confined animal feeding operations (purple), scaled by permitted number of animals allowed (CDPHE).

were conducted both in the GPMP shelter and in the National Park Service/Colorado State University Mobile Air Quality Laboratory stationed at the site.

## High-time resolution $NH_3$ and $NO_y$

High-time resolution ($\sim$5 s) measurements of $NH_3$ were made using a Picarro G1103 Analyzer, a cavity ring-down spectroscopy instrument, located in mobile laboratory. The inlet line (Teflon tubing) was located approximately 6 m above ground level (agl). The entire length of the inlet line was insulated and heated to 40 °C to minimize wall losses. A filter (Picarro P/N S1021) was placed on the end of the inlet to prevent particles from entering the instrument. Ammonium nitrate captured by the filter likely volatilized on the heated filter during sampling and is a potential source of bias in the $NH_3$ measurement. However, concurrent 24-h averaged measurements from an annular denuder/filter-pack sampler (see below) suggest that, assuming that all of the $NH_4NO_3$ sublimates and is measured as $NH_3$, the median bias in the measurements was 7.7%. A correction was not applied to the final dataset. Calibrations were performed at the beginning and end of the field deployment with a known concentration of ammonia, generated using a calibration gas diluted with a Teledyne Zero Air Generator (Model 701). A phosphorus acid (10%

w/v) coated denuder operated in parallel confirmed the supplied $NH_3$ concentration. The mobile lab, which housed the instrument, had an air conditioning failure July 23–30 resulting in conditions that were too hot for proper operation of the Picarro. After A/C was restored, a section of the heated inlet overheated and this problem was not discovered until August 8. Thus $NH_3$ data for the period July 23–August 9 are omitted from this analysis.

CASTNET $NO_y$ and NO data were downloaded from http://www.epa.gov/castnet (*US Environmental Protection Agency Clean Air Markets Division, 2016a*). A Teledyne API T200U/$NO_y$ instrument is operated at the ROMO CASTNET site (ROM206). This instrument measures both $NO_y$ and NO by chemiluminescence. NO is directly measured and $NO_y$ is converted using a thermal catalytic converter and measured as NO (*US EPA, 2017*). This method has an uncertainty of $\leq \pm 10\%$ with the additional caveats that some reduced organic nitrogen compounds may be converted and the conversion of the different $NO_y$ species is highly temperature dependent (*Crosley, 1996*; *Kondo et al., 1997*; *Williams et al., 1998*). The site is operated in accordance with the Quality Assurance Plan for Procuring, Installing, and Operating NCore Air Monitoring Equipment at CASTNET sites (*Amec Foster Wheeler, 2016*).

## High-time resolution organic compounds

An *in situ* gas chromatography (GC) system was set up in the mobile laboratory located at the ROMO site to measure hourly concentrations of select VOCs. This is a cryogen-free concentration and analysis system consisting of two GCs coupled to four columns. A VF-1701 (Varian Inc.) was used with an electron capture detector (ECD) for C1–C2 halocarbons and C1–C5 alkylnitrates; a Flame Ionization Detector (FID) was used with the three other columns: CP-$Al_2O_3$/$Na_2SO_4$ PLOT for detection of $C_2$–$C_7$ hydrocarbons, VF-1ms for $C_6$–$C_{10}$ hydrocarbons and a PoraBond Q column for selected oxygenated VOCs (OVOCs). The system, calibrated daily, is similar to those used in other studies (*Sive et al., 2005*; *Abeleira et al., 2017*). The GC was located in the mobile lab, data are unavailable for the July 23–30 period of the A/C failure.

Peroxyacetyl nitrate (PAN) was measured at ROMO using a custom gas chromatograph with 15-min time-resolution in the GPMP shelter. For the ROMO 2014 deployment, this custom system was configured to measure PAN, but not its homologues. The instrument operation and calibration are described in *Zaragoza et al. (2017)*. Briefly, the system used a ThermQuest ECD held at 50 °C and calibrations were performed regularly throughout the campaign using a photolysis cell to generate PAN with accurately measured flows of acetone in ultra high purity zero air and NO in nitrogen (Scott-Marrin Cylinder Numbers: CB09819 and CB11156).

## 24-h inorganic gases and particles

24-h inorganic gas and $PM_{2.5}$ particle chemistry was measured using URG annular denuders and filter-packs (URG Corporation, Chapel Hill, NC), replaced daily at 9:00 MDT, with a nominal volumetric flow of 10 L/min. These samplers were located outside the mobile laboratory at the ROMO site. Details on sample extraction and analysis can be found in previous work (*Beem et al., 2010*; *Benedict et al., 2013b*; *Benedict et al., 2013c*). The collected

samples were analyzed to determine the concentrations of gas phase $NH_3$ and $HNO_3$ and particle phase $NH_4^+$, $NO_3^-$, $SO_4^{2-}$, $K^+$, $Na^+$, $Cl^-$, $Mg^{2+}$, and $Ca^{2+}$.

## Wet deposition

Precipitation samples were collected using an automated NCON Systems (Arnoldsville, GA, USA) ADS/NTN Atmospheric Precipitation Sampler. Sample buckets were changed every two to three days depending on the site visit schedule. Sample buckets were cleaned with deionized water, shaken to remove water, covered with Al foil, and allowed to dry. Blanks were taken by pipetting 30 mL of high-purity deionized water into the bucket. Each bucket was weighed prior to installation in the field and upon returning to the lab to determine the total sample volume collected. Sample pH was measured after returning to the lab using a combination pH electrode and the remaining sample was frozen until analysis. Samples were analyzed for ionic composition ($NH_4^+$, $NO_3^-$, $SO_4^{2-}$, $Na^+$, $K^+$, $Mg^{2+}$, $Ca^{2+}$, and $Cl^-$) using ion chromatography (Dionex) with conductivity detection, and for total nitrogen by high temperature catalyzed combustion with chemiluminescent analysis of NO using a Shimadzu TOC $V_{CSH}$ + TNM-1 total organic carbon analyzer with total nitrogen module (*Benedict et al., 2013a*).

# RESULTS AND DISCUSSION

## Overview of sampling period

A timeline of the FRAPPÉ period is shown in Fig. 2 for the nitrogen species measured at high-time resolution, key tracer species, wind direction and precipitation. From the wind direction data, we can see that consecutive days of upslope flow (shaded region, 90–180°) from the Front Range, when winds are from the southeast, mainly occurred at the end of July and beginning of August and then again mid-month in August. We note that because the site is located in a valley, winds are channeled primarily in two directions (Fig. 3): from the southeast (upslope flow from the Colorado Front Range and northeast Colorado) and from the northwest (from less developed areas west of the Continental Divide). From previous work we know transport from the Front Range is an important source of nitrogen to ROMO (*Beem et al., 2010*; *Benedict et al., 2013c*); during periods of upslope flow in 2014 we also observe increases in the Front Range tracers and RN species. The pollution rose for PAN is shown as an example (Fig. 3) of the higher concentrations of RN species associated with upslope flow. This pattern of higher concentrations being associated with southeast (upslope) flow is similar for all of the RN species measured. On August 11 and 12 there were strong upslope winds and $NH_3$ was the highest observed during the FRAPPE study period. Others have suggested a fire influence in the Front Range during the August 11 and 12 period (*Dingle et al., 2016*), which may also contribute to RN concentrations (*Benedict et al., 2017*). Other upslope periods (e.g., July 22) show stronger signals in tracer species compared to $NH_3$. Throughout the study period, there appear to be strong correlations between the peak concentrations of PAN, R-ONO$_2$, ethane, and $C_2Cl_4$, suggesting similar sources and/or source regions, and $NO_y$ concentration timelines share similarities to PAN. Unfortunately, due to air conditioner failures, limited data were collected for portions of the study period.

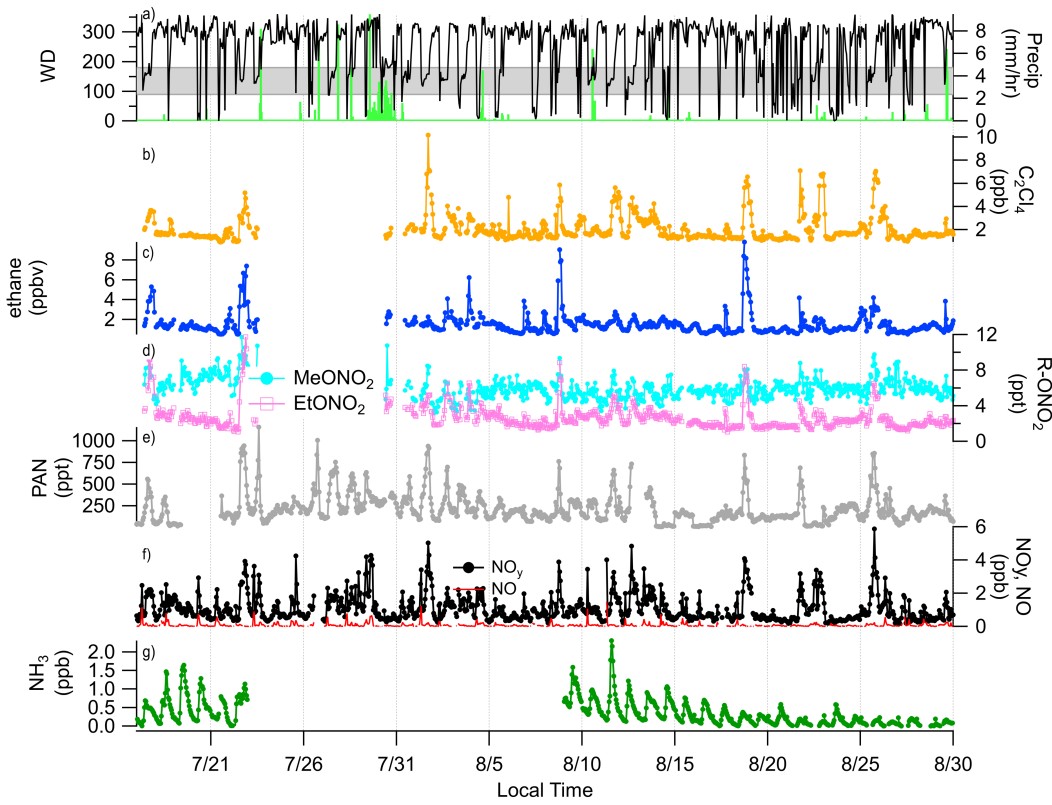

**Figure 2** **Timelines of reactive nitrogen and tracer species from July 16 to Sep 1, 2014 with wind direction and precipitation.** In (A) wind direction and precipitation are shown, the shaded region represented upslope flow at the sampling site. (B) $C_2Cl_4$; urban tracer, (C) ethane; oil and gas tracer, (D) $MeONO_2$ and $EtONO_2$, (E) PAN, (F) $NO_y$ and NO, (G) $NH_3$.

To improve our understanding of the sources and relationships between compounds, we next examine the diurnal cycles of several key N species (Fig. 4). Most of the species have a peak later in the day, likely associated with radiatively driven upslope transport, while there is an early peak in ammonia which is more likely to be associated with bi-directional exchange and the role of dew (*Wentworth et al., 2016*). An early morning peak in ammonia has been observed at other sites, including in Grand Teton National Park (*Prenni et al., 2014*), suggesting this may be a common occurrence. The early morning high concentrations of ammonia reflect local environmental effects (volatilization from evaporating dew) that strongly alter morning ammonia concentrations. There is no clear early morning peak of $NH_3$ observed in the August diurnal averages shown in Fig. 4, perhaps due to drier conditions. Earlier in the study (July) there were several large early morning peaks that are likely related to dew evaporation. To eliminate this factor from our source analysis, the morning (6 AM–10 AM) ammonia data were removed from the remainder of the analyses as any local factors that cause high $NH_3$ concentration would confound any interpretation of correlations with other species. $NO_y$ peaks early, between 6AM and 9AM, likely related to local morning traffic and then peaks again mid-afternoon through the evening with a

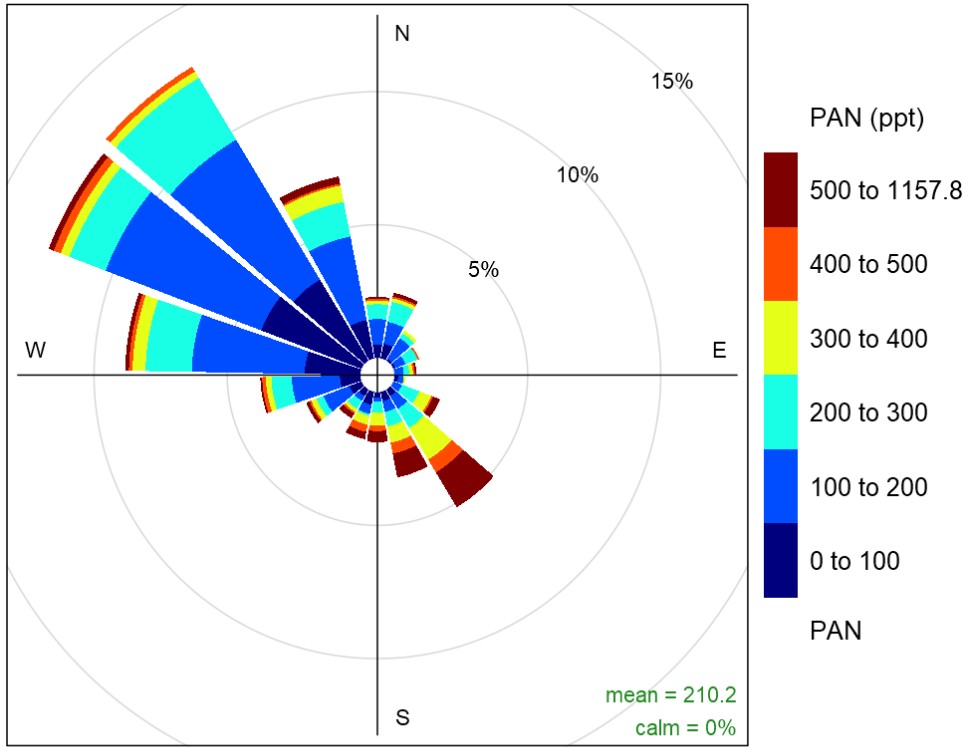

**Frequency of counts by wind direction (%)**

**Figure 3 Frequency of PAN concentrations by wind direction.** Distance from the center represents the fraction of time winds were from a given 20° wind sector and the colors in each paddle indicate the relative amounts of time concentrations of PAN were in a given concentration range.

lot of variability in when the highest late-day concentration is observed. The alkyl nitrates have the latest peak in concentration and are most similar to the diurnal cycles of the two tracer species ($C_2Cl_4$ and ethane). Concentrations of total R-ONO$_2$ are consistently low from the early morning (2 AM) through noon with the highest concentrations observed between 5 PM and 11 PM. On average the increases in PAN begin around 7 AM, likely associated with sunrise and the initiation of photochemical reactions. PAN slowly increases throughout the day with its average peak concentration occurring around 6 PM before concentrations begin to decrease overnight.

## Atmospheric reactive nitrogen budget

The 2014 period of intensive measurements provided an opportunity to examine the contribution of individual nitrogen compounds to the total atmospheric RN budget. For the FRAPPÉ period (July 16–August 18), $NO_y$ species were the most abundant, contributing 67% to the total RN budget (Fig. 5).

Based on our speciated measurements, $NO_y$ was partitioned as follows: 24% as PAN, 7% as NO, 4% as alkyl nitrates, and 11% as $HNO_3$ and $NO_3^-$. We estimate that the remaining 54% was $NO_2$ and other unmeasured oxidized nitrogen species ($NO_z$). We
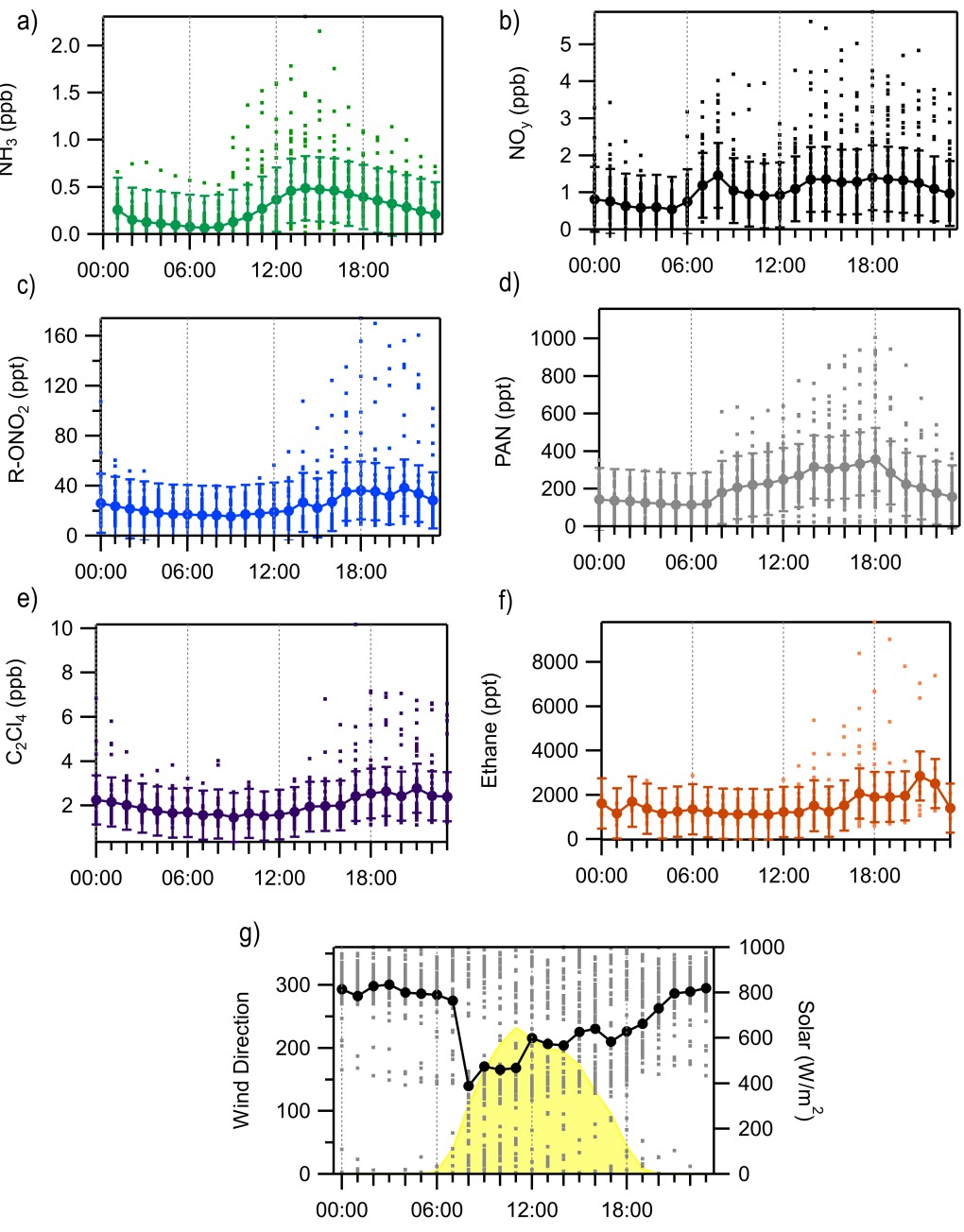

**Figure 4** **Diurnal averages and standard deviation of key RN and tracer species measured at ROMO from August 9 to August 31 with each valid measurement from every hour shown at the dots.** (A) NH₃, (B) NO$_y$, (C) sum of alkyl nitrates, (R-ONO₂), (D) PAN, (E) C₂Cl₄, (F) ethane, (G) wind direction and solar radiation.

note that we cannot separate NO₂ concentrations from other NO$_z$ contributions using the chemiluminescence instrument and converter system employed during FRAPPÉ. Other potential contributors to the NO$_z$ fraction include other organo-nitrogen compounds like hydroxynitrates, isoprene nitrates, and peroxyacyl nitrates other than PAN, but their

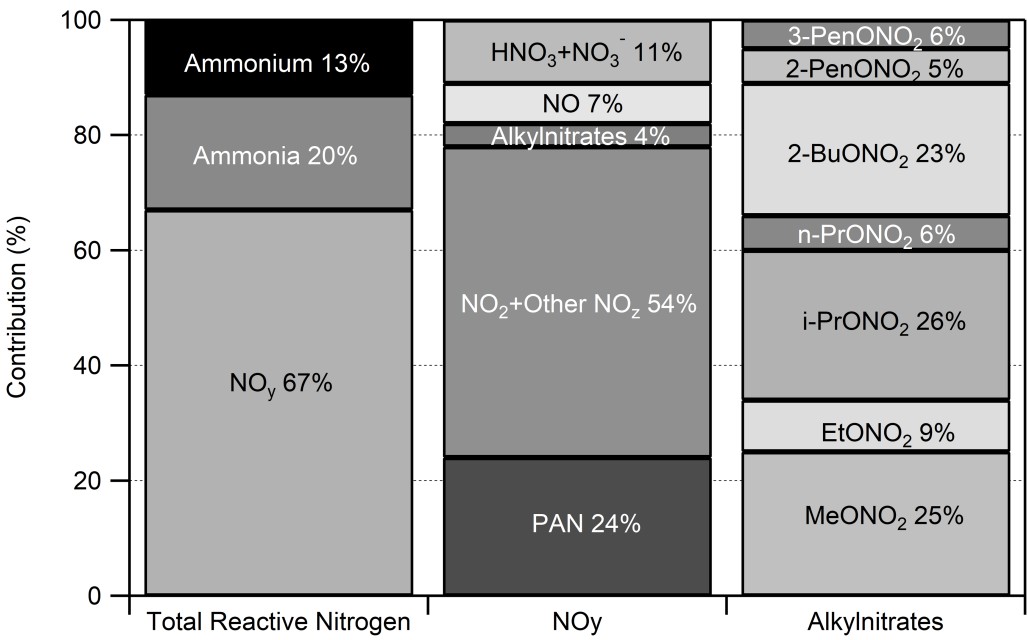

**Figure 5 Atmospheric reactive nitrogen budget at ROMO for July 16–August 31 2014.** Concentrations were averaged over the study period and the average total reactive nitrogen concentration was 1.24 ppb. Not included in this assessment are some basic, gas-phase reactive nitrogen species such as amines and acetonitrile.

importance is uncertain and is likely to vary spatially and temporally (*Parrish et al., 1993*; *Thornberry et al., 2001*; *Day et al., 2003*). At night, $NO_3^\bullet$ and $N_2O_5$ could be important contributors to $NO_y$ but in the presence of isoprene and unsaturated alkenes are likely to react, photolyze to $NO_2$ at sunrise, or, in the case of $N_2O_5$, undergo hydrolysis on aerosols to form particulate $NO_3^-$ (*Brown et al., 2004*; *Brown et al., 2006*). Previous measurements in Boulder and Denver show large temporal variability with concentration ranging from below detection to 100 pptv for $NO_3^\bullet$ and below detection to 3 ppbv for $N_2O_5$ (*Brown & Stutz, 2012*). While the specific contribution to nitrogen deposition is unknown for these species, they are expected to be captured in the $NO_y$ measurement.

A subset of the RN species have been measured at ROMO in previous years. Summary data from these measurements are given in Table 1. The overall partitioning between reduced and oxidized RN (i.e., leftmost column of Fig. 5) is consistent with measurements from other years (2009, 2010, 2015, 2016), when only $NH_3$, $NH_4^+$, $HNO_3$, $NO_3^-$, and $NO_y$ were measured. The four other study years measured $NO_y$, $NH_3$ and $NH_4^+$ from July through October. In this 2014 FRAPPÉ study the total RN was 1.6 ppb compared to 1.9–2.6 ppb in the other four studies.

Alkyl nitrates were measured for the first time in ROMO during the 2014 FRAPPÉ study. The alkyl nitrates measured here are primarily formed by oxidation of hydrocarbons emitted from fossil fuels (ethane, butane, pentane) and generally contribute less than 10% to $NO_y$ in continental regions (*Russo et al., 2010a*). Measurements of alkyl nitrates during FRAPPÉ were consistent with this, contributing 4% to $NO_y$ and 2.7% to the total atmospheric RN

**Table 1** July to August average reactive nitrogen concentrations (ppb) and the $NH_3$ and $NH_4^+$ fraction for the total oxidized and reduced nitrogen for multiple years of measurements made at RMNP. The FRAPPÉ study year is highlighted in bold for ease of comparison.

| | $NH_3$ | $NH_4^+$ | $HNO_3$ | $NO_3^-$ | $NO_y$ | $NH_4^+$ fraction | $NH_3$ fraction |
|---|---|---|---|---|---|---|---|
| 2009 | 0.47 | 0.37 | 0.10 | 0.07 | 1.74 | 14% | 18% |
| 2010 | 0.38 | 0.32 | 0.09 | 0.05 | 1.21 | 17% | 20% |
| **2014** | **0.34** | **0.20** | **0.06** | **0.03** | **1.05** | **12%** | **21%** |
| 2015 | 0.43 | 0.21 | 0.10 | 0.04 | 1.36 | 11% | 22% |
| 2016 | 0.58 | 0.25 | 0.07 | 0.03 | 1.22 | 12% | 28% |

budget during FRAPPÉ (Fig. 5). The measured alkyl nitrates were primarily composed of i-propyl nitrate (26%), methyl nitrate (25%), and 2-butyl nitrate (23%). The remaining 26% were ethyl nitrate (9%), n-propyl nitrate (6%), 2-pentyl nitrate (5%), and 3-pentyl nitrate (6%). All of the alkyl nitrate concentrations were highly correlated with each other, except methyl nitrate (Fig. 6). Whereas all of the other alkyl nitrates measured are primarily photochemical oxidation products, methyl nitrate ($MeONO_2$) has both a photochemical and marine source. To examine the influence of a potential marine influence, we examined the relationship between ethyl nitrate ($EtONO_2$) and $MeONO_2$ colored by $CHBr_3$ (Fig. 6B), a tracer for marine air (*Gschwend, MacFarlane & Newman, 1985*; *Carpenter et al., 2005*). As seen in the figure, $MeONO_2$ is enhanced during periods with elevated $CHBr_3$, suggesting that, although ROMO is relatively distant from marine sources, the park can be measurably influenced by marine air. Previous results from HYSPLIT back trajectories for the same site showed that air masses from the Pacific Ocean make it to ROMO (*Gebhart et al., 2014*), but this is the first chemical evidence of marine air masses at ROMO. Other possible contributing factors for the low correlation between $MeONO_2$ and the other alkyl nitrates may be the relatively higher solubility (faster scavenging by precipitation) and longer lifetime (oxidation and photolysis) of $MeONO_2$ (*Russo et al., 2010a*).

## Nitrogen deposition

Atmospheric nitrogen can be deposited to the surface of the earth by wet and dry deposition. Nitrogen in wet deposition comes from activation of cloud condensation nuclei (CCN) and subsequent scavenging of gases and non-activated aerosols by cloud droplets, rain, and snow. Dry deposition is direct deposition of particles and gases to the surface. Dry deposition is more difficult to quantify than wet deposition because it is difficult to measure directly, and modeled estimates can depend on chemistry and environmental factors including relative humidity, temperature, atmospheric turbulence, boundary layer thickness, and the surface to which deposition occurs (*Tarnay et al., 2001*).

## Wet deposition

Consistent with previous ROMO studies (*Beem et al., 2010*; *Benedict et al., 2013a*) wet deposition dominated the total nitrogen deposition budget during FRAPPÉ, contributing 0.79 kg N/ha or 85% to total nitrogen deposition (Fig. 7). Ammonium contributed 49%, $NO_3^-$ contributed 34%, and organic nitrogen contributed only 2% to total nitrogen deposition for July 17–August 31. While the fraction of organic nitrogen measured in the

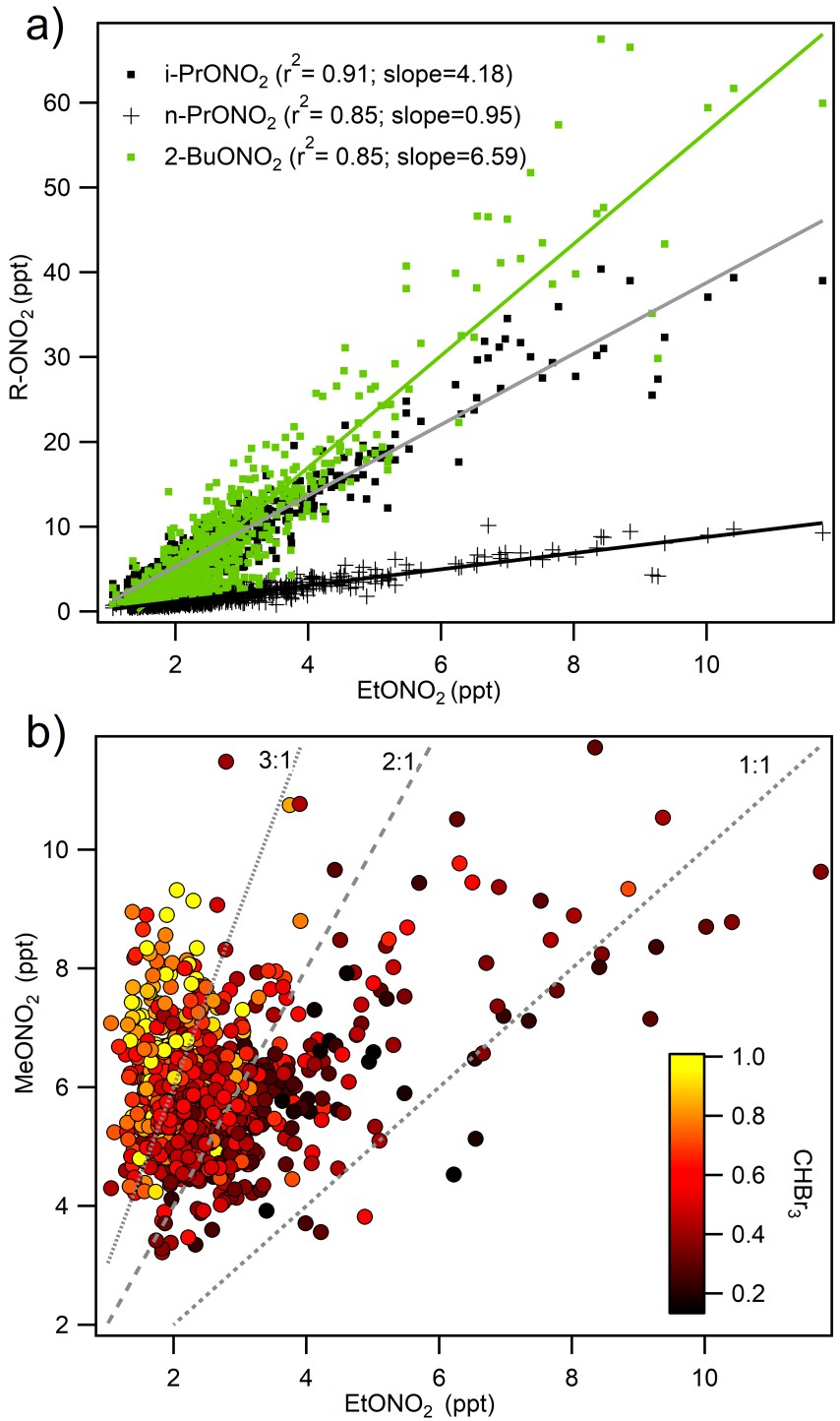

**Figure 6** **Relationship of methyl nitrate and other alkyl nitrates to ethyl nitrate.** (A) i-propyl, n-propyl, and 2-butyl nitrate shown with ethyl nitrate and (B) the relationship between methyl and ethyl nitrate colored by mixing ratios of CHBr$_3$ (ppt), a marine air mass tracer highlighting the two sources of methyl nitrate at ROMO.

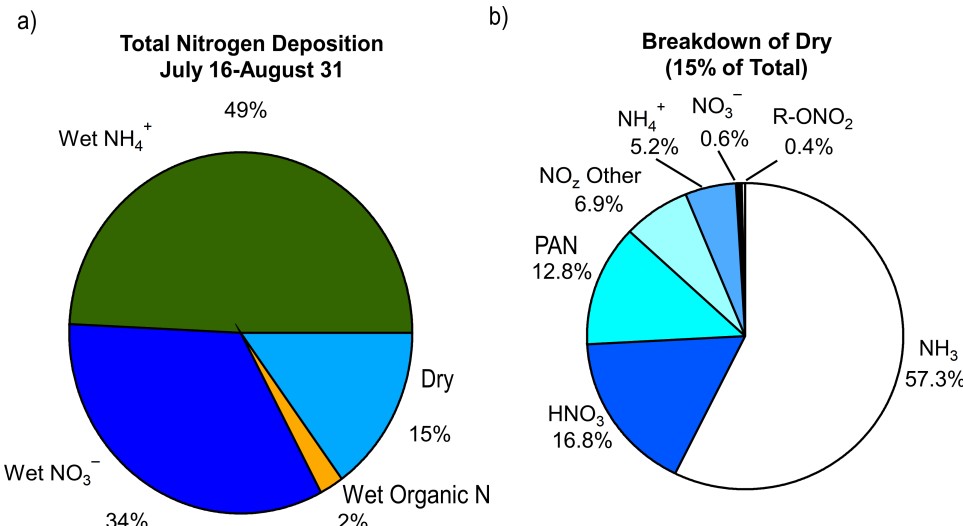

**Figure 7** Contributions of the (A) various reactive nitrogen species measured to total nitrogen deposition and (B) breakdown of dry deposition.

gas phase was much higher than 2% ($\sum$R- ONO$_{2(g)}$ + PAN$_{(g)}$ = 22%) this fraction is likely to be lower in precipitation due to the significantly lower solubility of the different organic nitrogen gases (relative to ammonia and nitric acid), and due to potential for loss in collected samples.

The fraction of organic nitrogen (ON) within individual events varied greatly in summer 2014, from 0% (ON below detection) to 20%. Although the average wet deposition of ON was less than we have previously observed at this location, the upper bound is consistent with measurements during the 2009 RoMANS study, when the average ON fraction in late summer and early fall was at 20% (*Benedict et al., 2013a*). Understanding the cause for the differences in the two study years is difficult, because we quantify bulk organic nitrogen and not individual species, which provides little information on source types.

At the measurement site in ROMO average precipitation from July to September for the period of 1995–2016 is 179 mm. Annually the changes can be quite large; the standard deviation is 98 mm. Table 2 summarizes the precipitation and wet nitrogen deposition for 2009, 2010, and 2014–2016. In 2014 total precipitation from July to September was well above average (222 mm) and a large fraction of it (63 mm or 32%) fell during upslope wind events (surface winds 70–180°), resulting in high wet deposition values. This is a consistent trend we observed at the ROMO site, years with high precipitation, especially large upslope precipitation fractions tend to have the highest wet N deposition (for example 2009). For the years we have data, 2015 is a case with a lower fraction of precipitation occurring during upslope flow conditions and the lowest wet N deposition likely due to variations in synoptic and local circulation. The variability in precipitation patterns is an important driver in the total N deposition observed due to the importance of wet deposition at the site. Additionally, the fraction of reduced nitrogen or oxidized in the wet deposition do not show any trend with the amount or fraction of precipitation associated with upslope flow,

**Table 2  Summary of precipitation for July–September for the intensive study years in ROMO.** Wet and dry nitrogen deposition are also included for comparison.

| Year | Precipitation | | | Fraction of time flow upslope | Wet N deposition July–Oct (kg N/ha) | Estimated dry NH$_3$, HNO$_3$, NH$_4^+$, NO$_-$ deposition (kg N/ha)[a] |
|------|------------------------|------------------|------------------------------|--------|------|------|
| | During upslope (mm) | Total 7/1–9/30 | Fraction during upslope | | | |
| 2009 | 32.6 | 102 | 33% | 33% | 1.24 | 0.38 |
| 2010[b] | 39.9 | 170.9 | 26% | 23% | 0.82 | 0.25 |
| 2014[c] | 62.8 | 221.8 | 32% | 37% | 1.37 | 0.21 |
| 2015 | 17.1 | 108.1 | 18% | 25% | 0.72 | 0.28 |
| 2016 | 23.3 | 60.4 | 30% | 27% | 0.85 | 0.35 |

**Notes.**
[a]Dry ammonia deposition was calculated from daily NH$_3$ from denuder measurements to keep the calculation consistent across all study years.
[b]July 1–Sept 20.
[c]July 7–Sept 30.

or the fraction of time upslope occurred. Specifically, the largest reduced fraction (66%) was measured in 2016 and the smallest (46%) in 2009 and neither of these years stand out in the parameters we've examined in Table 2.

## Dry deposition

For estimating dry deposition during FRAPPÉ, we used a multi-faceted approach, depending on the data. For daily data, we used the monthly average diurnal cycle of deposition velocity ($V_d$) from CASTNET (*US Environmental Protection Agency Clean Air Markets Division, 2016b*) to determine $V_d$ for HNO$_3$, NO$_3^-$, and NH$_4^+$, and then applied these hourly $V_d$ values to the 24 h average concentrations observed (average $V_d$(HNO$_3$) = 1.8 cm/s, $V_d$(PM$_{2.5}$) = 0.20 cm/s). The diurnal cycle for $V_d$ was based on monthly average hourly values of all available deposition velocities at the ROMO site (1/1995–1/2014). For NH$_3$, which is measured at higher time resolution (1-h), hourly averages were taken from historical data (1/1995–1/2014) for $V_d$(HNO$_3$) and were scaled by 0.7 based on a literature survey of co-located HNO$_3$ and NH$_3$ $V_d$ measurements (*Benedict et al., 2013b*; *Benedict et al., 2013a*) (average $V_d$(NH$_3$) = 1.25 cm/s). This may overestimate actual net deposition of NH$_3$ given the importance of bi-directional exchange (*Li, 2016*; *Wentworth et al., 2016*). Finally, for other gas phase nitrogen species that are not measured or modeled by CASTNET, we examined literature values of fluxes and deposition velocities to estimate their contributions to the nitrogen deposition budget during FRAPPÉ, as discussed below.

Estimating dry deposition of NO$_y$ is difficult due to the variability in NO$_y$ composition across study sites and at different times of year. To that end we have estimated the dry deposition of the separate components measured. Hourly deposition velocities for PAN were taken from *Turnipseed et al. (2006)* for a Loblolly pine forest in North Carolina (daily average 0.52 cm/s). However, we note that *Wolfe et al. (2009)* published hourly values of PAN deposition velocities for a Ponderosa pine forest in the Sierra Nevada mountains that are significantly lower than Turnipseed values (average = 0.17 cm/s). We use the higher Turnipseed values to provide an upper limit for deposition for PAN. This corresponds to a factor of ~2.7 difference in the total dry deposition of PAN based on these estimates

of $V_d$. Dry deposition of alkyl nitrates was estimated using the nighttime $V_d$ (0.13 cm/s) from *Russo et al. (2010a)*, one of the few published studies that estimates an alkyl nitrate deposition velocity. The contribution of dry NO deposition to total N deposition has been neglected, as atmospheric chemical reaction is the primary means of removal in the lower atmosphere due to the low solubility and oxidizing capacity of NO (*Wesely & Hicks, 2000*) and low NO concentrations. In order to provide some estimate of fluxes of the unspeciated components of $NO_z$, dry deposition was calculated using the average hourly and monthly (1/1995–1/2014) $O_3$ deposition velocities from CASTNET (*Clarke, Edgerton & Martin, 1997*; *Schwede et al., 2011*) scaled by 0.67 (*Wesely & Hicks, 2000*) (average $V_d(NO_z) = 0.12$ cm/s).

Estimated dry deposition contributed 0.14 kg N/ha, or 15%, of total deposition in ROMO during the FRAPPÉ period. Dry deposition was dominated by $NH_3$ (57%; 0.08 kgN/ha) followed by $HNO_3$ (17%; 0.024 kgN/ha), PAN (13%; 0.018 kgN/ha), and $NO_z$ (7%; 0.01 kgN/ha). Of the dry deposited species, $NH_3$ contributes the most to total deposition (8.6%). Species that are newly measured in this study have small contributions to total nitrogen deposition, including PAN (2%), $NO_z$ (1%), and alkyl nitrates (0.1%; 0.005 kgN/ha), which are of similar importance to $HNO_3$ (2.5%), $NH_4^+$ (0.8%) and $NO_3^-$ (0.1%) dry deposition.

Wet nitrogen deposition is directly measured with ∼10% uncertainty and should be used as a lower bound of total nitrogen deposition. Estimating the dry component of the nitrogen deposition budget is fairly uncertain but does provide some idea of its potential importance. Limitations of the method to determine dry deposition used here include using average $V_d$ from the site instead of values for the meteorological conditions during the study, using non-site specific values for PAN and $NO_y$, and not accounting for bi-directional exchange of $NH_3$ but using the nitric acid $V_d$ ratio should account for some of this interaction. To truly get a handle on the dry deposition component, and the uncertainty in the estimates made here, direct flux measurements need to be made particularly since the dry components are not insignificant to the total estimated nitrogen deposition budget.

## Sources impacting ROMO

The VOC measurements at ROMO during FRAPPÉ provide insight into the array of potential sources and source regions that may contribute to excess nitrogen deposition in the park. In an analysis of the VOC data from FRAPPÉ, *Benedict et al. (2018)* observed the contribution of several sources impacting $O_3$ at ROMO, with some days showing more oil and gas influence, while others had an increased urban influence.

Here we focus on compounds that help separate urban pollutants from those most likely associated with oil and gas production. Tetrachloroethene ($C_2Cl_4$) has been used extensively as a tracer for urban air (*Simpson et al., 2004*) because it is used as a dry cleaning solvent and degreasing agent. Ethane is mainly emitted to the atmosphere during the production, processing and transportation of natural gas (*Tzompa-Sosa et al., 2017*). Table 3 displays the correlation ($R^2$-values) between the measured RN species and these two VOC tracers. A significant fraction of the observed variability in $NO_y$, PAN, and the

**Table 3  Correlation ($R^2$) between the tracer species (ethane, $C_2Cl_4$) and the various RN species (ppb).**

|  | Ethane ($R^2$) | $C_2Cl_4$ ($R^2$) |
|---|---|---|
| $NH_3$ | 0.01 | 0.01 |
| $NO_y$ | 0.33 | 0.59 |
| NO | 0.003 | 2E-4 |
| PAN | 0.27 | 0.35 |
| $MeONO_2$ | 0.05 | 0.05 |
| $EtONO_2$ | 0.68 | 0.42 |
| $i$-$PrONO_2$ | 0.79 | 0.47 |
| $2$-$BuONO_2$ | 0.85 | 0.44 |
| $2$-$PenONO_2$ | 0.87 | 0.42 |
| $3$-$PenONO_2$ | 0.86 | 0.44 |

alkyl nitrates (except $MeONO_2$) can be explained by the variability in these tracers. $NO_y$ is more strongly correlated with the urban tracer, $C_2Cl_4$, while the alkyl nitrates are more strongly correlated with the oil and gas tracer, ethane. PAN is not strongly correlated with either tracer likely because both sources types (oil and gas/urban), long range transport, and photochemical processes contribute to observed concentrations. The strong correlation between alkyl nitrates and ethane is expected, since the parent compounds of the alkyl nitrates (e.g., propane, butane, and pentane) come from similar sources as ethane (*Roberts, 1990*). No significant increase in the correlation with any of the RN compounds is observed when moving from a single regression to a multi-linear regression using ethane and $C_2Cl_4$ (not shown).

The ratio of i-butane to n-butane has been used extensively to indicate the relative importance of different VOC source types to an air parcel; low values of this ratio can correspond to ''wet'' oil and gas production or transport (0.37 Marcellus Shale) (*Swarthout et al., 2015*) and higher values of this ratio indicate rural air masses (0.52 New Hampshire) (*Russo et al., 2010b*) or ''dry'' oil and gas production (1.07 Piceance, 0.685 Uintah) (*Helmig et al., 2014*; *Hilliard, 2016*). In Fig. 8 the tracer and nitrogen-containing compounds are plotted against the i- to n-butane ratio to show how they align with the air masses measured in Denver during FRAPPÉ (0.45) and a regional Denver-Julesburg (DJ) Basin value measured at the BAO tower (0.40) (*Swarthout et al., 2013*). Individual points are colored by wind direction to provide some information on the transport of the associated air masses and it is important to note, there is a strong relationship between the i- to n-butane ratio with wind direction. For ethane, the highest mixing ratios were observed when the i- to n-butane ratios were near 0.4, consistent with those observed in the DJ Basin. There was significantly more scatter observed for mixing ratios <2 ppb, suggesting mixing of air masses that have been transported over a longer distance. Elevated mixing ratios of PAN and $NO_y$ were associated with i- to n-butane ratios near both Denver and the DJ Basin, while elevated mixing ratios of the alkylnitrates, which are primarily oxidation products of the light alkanes, were observed when i- to n-butane ratios matched those observed within the nearby DJ basin. For $NH_3$, there was a lot of scatter in i- to n-butane

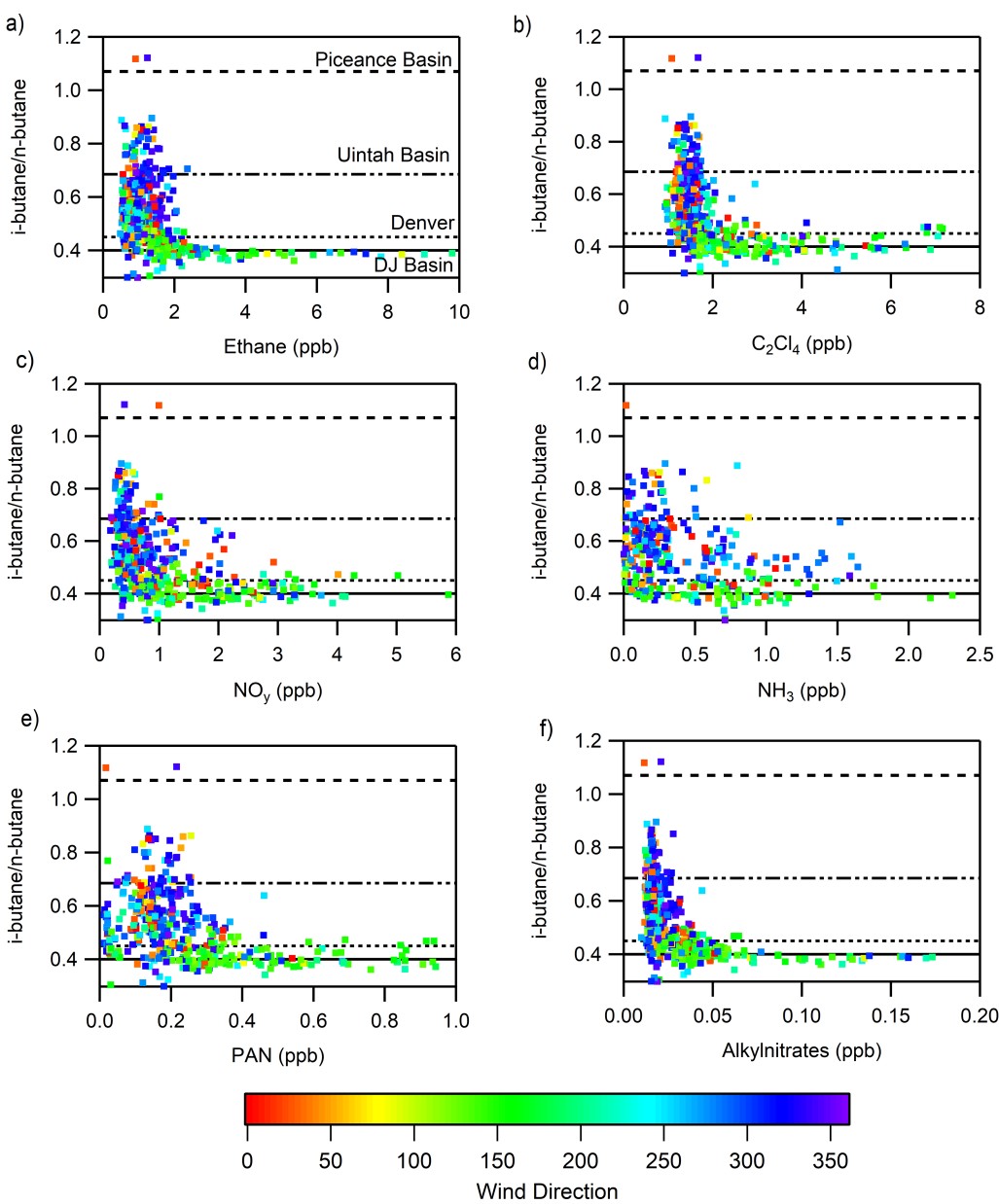

**Figure 8** **Relationships of the ratio of i-butane to n-butane to various species to separate the influence of Denver (solid) and the agriculture/oil and gas region (dotted) in the Front Range.** The i-butane to n-butane ratio and its relationship to (A) ethane, (B) $C_2Cl_4$, (C) $NH_3$, (D) $NO_y$, (E) alkynitrates, (F) PAN are shown.

ratios at mixing ratios below 1.5 ppb; however, higher mixing ratios were associated with i-to n-butane ratios similar to those observed within the nearby DJ basin. This likely results due to oil and gas operations being collocated with extensive agricultural operations in this basin.

If we extend the analysis from Fig. 6 where we observed the influence of long range transport using high concentrations of $CHBr_3$ and ratios of $MeONO_2/EtONO_2 > 3$

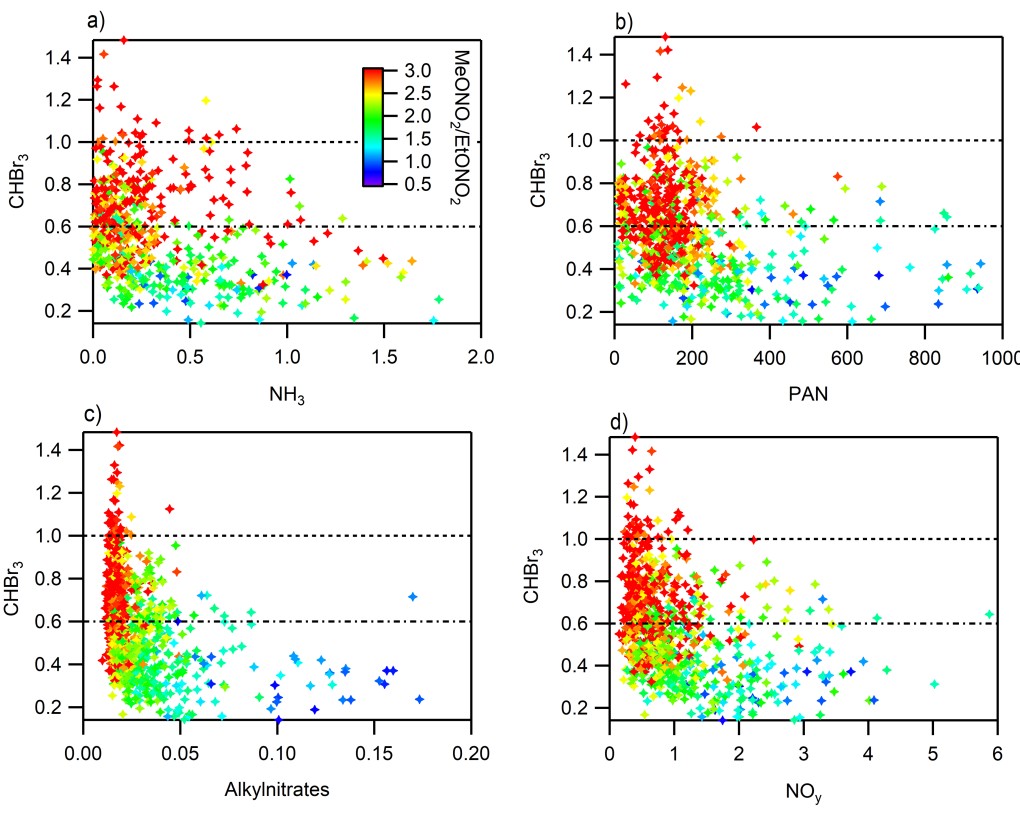

**Figure 9** **Relationship between reactive nitrogen species and CHBr₃ colored by MeONO₂/EtONO₂.** All points with a ratio greater than 3 are red. (A) NH₃, (B) PAN, (C) sum of alkyl nitrates, (D) NO_y.

to the RN species we observe that in some cases the RN species are elevated but not significantly (Fig. 9). The concentrations of alkylnitrates and NO_y during periods when long range transport was most likely to occur, CHBr₃ > 1 ppbv and higher ratios of MeONO₂/EtONO₂, are low and near zero. PAN shows slight elevation during these conditions which is expected due to the greater potential for long range transport at high altitudes and cooler temperatures. However, the majority of high PAN concentrations were not observed under these conditions. Ammonia shows a different pattern with more spread in the concentrations observed at higher CHBr₃, some of this variability may be due to local CHBr₃ emissions associated with a waste water treatment plant but there are few data available that examine CHBr₃ emissions from this type of source (*Gschwend, MacFarlane & Newman, 1985*; *Carpenter et al., 2005*). Using this metric is imperfect due to the potential for other sources to contribute to elevated CHBr₃, but still the observations with the high ratio of MeONO₂/EtONO₂ are more likely to be from long range transport while the other points are more likely to be associated with transport from the Front Range.

## CONCLUSIONS

Measurements made at ROMO during the FRAPPÉ study in 2014 provided an opportunity to add measurements of RN that had not been previously made at the site (PAN, alkyl

nitrates) and to try to connect observed species and their abundances with specific tracer species to better understand the source regions. In the atmospheric RN budget the relative importance of urban/combustion and rural/agricultural influence is often viewed as the oxidized/reduced nitrogen ratio; however, the rural eastern plains of Colorado are strongly influenced by both agriculture and oil and gas operations, the latter offering significant $NO_x$ emissions (*Duncan et al., 2016*). So while RN is dominated by oxidized nitrogen in the gas phase, it is not clear that urban air is the dominant contributor of RN deposition to ROMO. In deposition, reduced nitrogen is the major component. The differing importance of reduced (more in precipitation) and oxidized (more in the atmosphere) nitrogen species in the gas and the aqueous phase is a reflection of the different solubilities and abilities to be scavenged based on the properties of the individual species. It also suggests that emissions from many types of sources are important for the observed air mass composition and deposition at ROMO.

In this study we observed evidence that emissions from oil and gas operations, urban sources, and long range transport impact ROMO and affect atmospheric RN. The use of VOC tracers confirmed previous measurement (*Benedict et al., 2013c*) and modeling (*Gebhart et al., 2011*; *Thompson et al., 2015*) studies that identified these sources as important in the region. The spatial overlap of agricultural and oil and gas operation sources in eastern Colorado indicate agriculture is also an important source of RN even though we did not use a specific tracer for the source. Other measurements of ammonia and the other RN species, beyond this study, were made during FRAPPÉ. Specifically, there are mobile and aircraft measurements that should be analyzed to determine if more information about specific sources in NE Colorado can be identified. Further analysis of the extensive FRAPPÉ dataset is beyond the scope of the current study. However, it would be beneficial to do a more comprehensive analysis of the FRAPPÉ aircraft data using the ROMO site as a point of reference to better understand the spatial variability of these species.

Previous nitrogen deposition budgets constructed for ROMO lacked $NO_y$ and gas phase organic nitrogen compounds as part of the dry deposition budget (*Beem et al., 2010*; *Benedict et al., 2013a*). As seen in previous work, the nitrogen deposition budget at ROMO is dominated by wet deposition, specifically $NH_4^+$ followed by $NO_3^-$. Our estimates of dry deposition suggest that reduced nitrogen (specifically $NH_3$) is the largest contributor to the dry deposition budget. This is the first study in ROMO that includes deposition estimates of PAN (1.9% of total N deposition) and alkyl nitrates (0.06% of total N deposition). Although their contributions were relatively small, it is important to examine all components of the nitrogen deposition budget to better understand the role of various sources to nitrogen deposition and better constrain the deposition budget. Providing tighter constraints would require measurements of dry deposition, rather than the inferential estimates as done here. We also lack information on bi-directional fluxes, of $NH_3$ in particular, needed to better understand the deposition budget at ROMO.

# ACKNOWLEDGEMENTS

The assumptions, findings, conclusions, judgments, and views presented herein are those of the authors and should not be interpreted as necessarily representing the National Park Service. Meteorological data were provided by the National Park Service. We would like to recognize the contributions of the FRAPPÉ/Discover-AQ PIs (Gabriele Pfister-NCAR, Frank Flocke-NCAR, Jim Crawford, NASA) for their contributions in organizing and directing the experiment including the flight and measurement planning and field operations.

## Funding

Support for this work was provided by the National Park Service. The funders had no role in study design, data collection and analysis, decision to publish, or preparation of the manuscript.

## Grant Disclosures

The following grant information was disclosed by the authors:
National Park Service.

## Competing Interests

Jeffrey L. Collett Jr is an Academic Editor for PeerJ.

## Author Contributions

- Katherine B. Benedict and Anthony J. Prenni performed the experiments, analyzed the data, prepared figures and/or tables, authored or reviewed drafts of the paper, approved the final draft.
- Amy P. Sullivan, Ashley R. Evanoski-Cole, Emily V. Fischer, Sara Callahan and Yong Zhou performed the experiments, authored or reviewed drafts of the paper, approved the final draft.
- Barkley C. Sive, Bret A. Schichtel and Jeffrey L. Collett Jr conceived and designed the experiments, authored or reviewed drafts of the paper, approved the final draft.

## Data Availability

NASA Airborne Science Data for Atmospheric Composition, Discover-AQ/FRAPPE https://www-air.larc.nasa.gov/cgi-bin/ArcView/discover-aq.co-2014?C130=1.
Additionally, we have included all of the data used in this analysis as Supplemental Files.

## Supplemental Information

Supplemental information for this article can be found online at http://dx.doi.org/10.7717/peerj.4759#supplemental-information.

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
