# Peer review of "Impact of Front Range sources on reactive nitrogen concentrations and deposition in Rocky Mountain National Park"

_PeerJ, doi:10.7717/peerj.4759_

## Round 0.1 · original submission · Major Revisions

This paper is interesting and adds a good dataset of chemically speciated reactive nitrogen to the literature. I would reiterate the point made by some of the reviews, the metadata link to the data used in the paper does not take you to the actual data and this should be made clearer. I would also encourage you to address all of the points made in the reviews especially some of the points that indicate that the narrative of the data is not consistent with the graphical depiction of the data.

No work on the reactive nitrogen budget should fail to at least consider the contribution of dinitrogen pentoxide. Recent work is finding that nocturnal uptake of this molecule is significant source of PM nitrate. An atmospheric reactive nitrogen budget is not complete without at least some discussion and consideration of this important molecule. I would like to see at least some discussion and estimation/review of previous work on the likely contribution.

The paper also need to broader discussion of global picture with regard to reactive nitrogen budgets and whilst I realise that this is an analysis of regional US situation, it is reasonable to expect this work to be put into a global context and acknowledge the large body of work on reactive nitrogen that has been published, for example as part of the EU funded nitroeurope project.

Reviewer 1 ·

Basic reporting

The paper presents measurements of PAN, alkyl nitrates, NOx, NOy and NH3 from a site in Rocky Mountain National Park, Colorado, during the Front Range Air Pollution and Photochemistry Experiment (FRAPPE) in July-August 2014. The paper is generally well written and professionally presented. Sufficient background is provided for the reader to understand the importance and context of the measurements.

The link to the raw data directs you to the general FRAPPE data repository. Finding the specific data covered in this article is not easy (I managed to find it after some puzzled searching), so I recommend that a more specific link be provided that sends the reader to the specific dataset(s) that are reported here.

Experimental design

The work presented here is original primary research and adds to the general understanding of atmospheric reactive nitrogen concentrations and deposition. The methods used and data obtained are described in sufficient detail and demonstrate the authors' ability to perform this kind of research at a high technical and ethical level.

Validity of the findings

The findings of this research are well supported although they could be strengthened as described below. The analyses are appropriate and statistically sound. Although not earthshaking results, the data and analysis add to the general body of research of atmospheric reactive nitrogen.

The data in Figure 2 could be presented in a more powerful way by creating windrose plots for each of the species shown. The time dimension of the data would be lost but there doesn't seem to be a substantial temporal variation to the data anyway. Using windrose plots would effectively highlight how the species concentrations vary as a function of wind direction. As it is presented now, the reader has to do quite a bit of work to determine what each species concentrations are for a given wind direction (e.g., upslope flow from the southeast).

Additional comments

Editing Suggestions

Line 137: Be consistent with the format of dates … “July 23 – August 9” rather than 7/23 – 8/9.

Lines 194-197: It’s not at all clear what days/times on which upslope (from the southeast) flow occurs and so it’s very difficult to correlate the wind direction with species concentrations. The suggestion above was to create windrose plots of each species. However, if this is not feasible, at least modify the wind plot of Figure 2 to allow the reader to easily see which wind directions are “upslope” flow.

Line 207: “July 22” rather than “7/22”.

Line 289: Should be “raise” not ‘raising’.

Line 334: “We have” not “We’ve”, or better yet “The contribution of dry NO deposition to total N deposition has been neglected, as …”.

Line 366-368: Suggest the following … “No significant increase in the correlation with any of the RN compounds is observed when moving from a single regression to a multi-linear regression using ethane and C2Cl4.”

Line 439: Should be “… estimates of deposition of PAN …”.

Figure 7: The caption panel letterings do not correctly correspond to the figures as shown for (c) and (d), and (e) and (f).

Figure 8: A different color scale for the MeONO2/EtONO2 ratio might help to make this plot more easily interpreted. As it is presented now, the very dark color overwhelms the other colors.

·

Basic reporting

As I am not a native English speaker and I do not comment the language.

Experimental design

See General comments.

Validity of the findings

no comment

Additional comments

This is a compact MS about RN measurement and I have just minor comments:

Abstract: to A limited set of measurements was extended through October. ( this is a detail not for the abstract)

Indroduction: Information like the instruments and their time resolutions and details about the compounds measured should be written in the methods part, not in the introduction: ….high time resolution (1 min to 1 hour)… 24 h.. two gas chromatograps…..

Methods
Write more logically and think what is relevant: some methods have more details about where there were situated (like mobile lab?) or about the calibration (pH buffers 4 and 7). Is it possible to have a table of the methods?

Results
Check the %-values in the text, they are not in agreement with the figures (or are the figures correct):

r. 253- (Alkyl nitrate?) Measurements during FRAPPÉ were consistent with this, contributing 4% to NOy and 2% (I got 2.7% ?) to the total atmospheric RN budget during FRAPPÉ (Figure 4). The measured alkyl nitrates were primarily composed of i-propyl nitrate (26%), methyl nitrate (25%), and 2-butyl nitrate (23%). The remaining 26% were ethyl nitrate (9%), n-propyl nitrate (6%), 2-pentyl nitrate (6%) – in the figure 5%, and 3-pentyl nitrate (5%) – in the figure 6%.

r. 286 …than 2% (ΣR-ONO2 + PAN = 22%) – in the figure 6: 12.8% + 0.4 % ≠ 22%

It would be nice also to see clearly the deposition amounts
not only

Conclusions
r. 406 add FRAPPE 2014 (just easier to remember for the reader.)
r. 425-431 This part there is unclear speculation. Write this part more compact and clear. (other NH3 and RN measurements – which measurements – are beyond this study? Further analysis beyond of this study…)

Details:
Check the heading: sometimes like 2. Methods and sometimes 3.0 Results etc…
Figure 1 (map): add the scale.
r. 53 Elevated nitrogen deposition can alter ecosystems and the services they provide... services?
r. 99 NOy – not explaned
r. 117 NPS/CSU -not explaned?
r. 156-157 non-methane hydrocarbons (NHMCs) – Use NMHC or you do not need to mention this at all because there is already C2-C7 and C6-C10
r. 167 You do not need the abbreviation UHP, because you do not use it in the rest of manuscript.
r. 178 What is NCON? A company?
r. 183 Sample pH was measured after returning to the lab using a combination pH electrode calibrated with pH 7 and pH 4 buffers – too detailed or you should not write how you calibrated all the other instruments.
r. 206 Benedict et al., 2017 - Not mentioned in the reference list (there it is 2018?).
r. 289 fractions observed here raising – are raising?
r. 299 At the measurement site in ROMO average precipitation from July to September for the period of 1995-2016 is 179 mm – so this means the average precipitation for the 3 months period (Jul-Sep)
r. 318 hourly monthly averages?
r. 357 it is used a dry…
r. 439 This is the first study in ROMO that includes estimates deposition of PAN – estimated or estimates of PAN deposition..
r. 401 (Gschwend, MacFarlane & Newman, 1985; - should be
Gschwend et al. 1985

Figure 2. Precipitation: I have difficulties to see the green line close to the black one (WD).

Reviewer 3 ·

Basic reporting

In their manuscript, Benedict et al. describe measurements on reactive nitrogen (RN) species during a field study (i.e., FRAPPÉ) in summer 2014 in the Rocky Mountain National Park. In particular, they measured PANs, C1-C5 alkyl nitrates, NOx, NOy, and ammonia. Additionally, VOC measurements were conducted, using in situ GC-ECD and GC-FID. From these measurements, the authors conclude that concentrations of RN mostly correlate with emissions from nearby oil and gas operations as well as emissions from agriculture and urban areas. For some periods, long-range transport is also suggested as source of RN in the study area. Moreover, the authors calculate an atmospheric RN budget, including a nitrogen deposition budget for dry and wet removal processes.

The language quality is high and the author discuss their findings mostly in a clear way. Nonetheless, I would suggest shortening some rather long sentences and rephrasing certain sections, to improve the clarity of the manuscript (see general comments).
The article includes sufficient introduction and background and generally references relevant literature (see the general comments for some further suggestions).

All figures presented are relevant, high quality, and well described.

Raw data is not supplied with the manuscript and can ,therefore, not be assessed.

I would suggest publication of the manuscript after major revisions of the findings, as stated in the comments below.

Experimental design

The research question is well defined and the experimental setup and design appropriate to answer it. All methods applied are well described with sufficient detail to replicate such measurements.

Validity of the findings

Figure 1: The source of the data shown in this map remains unclear.
L207-211 / Figure 2: Please give more evidence to support your suggested correlation between PAN, RONO2, ethane, and C2Cl4. I would suggest depicting the relationships in a correlation plot rather than in a timeline plot. For the reader, it is hard to tell from a timeline if the signals really correlate or not.
Figure 3: Instead of showing every data point, I would suggest to show the standard deviation of the hourly averages in each panel. This would substantially improve the readability of the figure and make it easier to the reader to assess the total variability.
L216 / Figure 3: I cannot see a peak in ammonia for the early morning hours in Figure 3! From the depicted data, I would actually say that there is a peak between 2-4 PM. Thus, your suggested explanation that this peak reflects volatilization from evaporating dew cannot be correct.
L221: I do not understand why you remove all data for ammonia from 6-10 AM. Even if there was an early morning effect, it would not be a measurement artifact, but reflect actual RN concentrations in the air. This point is crucial, since it affects the entire data set and calculations of the study.
L245-249: If you state that “overall partitioning […] is consistent with previous years”, you have to show it either by giving references or by showing the data. Moreover, it remains unclear, which studies you are discussing when it is stated “All four studies […]” (L247) and “the other three studies” (L249).
L263: To me it seems very interesting that you can measure such long-range transport of marine tracers. However, marine sources are not the only possible explanation for the abundance of CHBr, as you also admit later in section 3.4 (see L399). Thus, I would strongly recommend adding backward trajectories (e.g., from HYSPLIT) to demonstrate the origin of the air masses arriving at the site.
L283: It would be interesting to know a bit more about the uncertainty range in these calculations.
L295-298: This (very long) sentence remains completely unclear to me. What message do you want to convey here? Which other studies do you mean?
L299-308: The discussion of the precipitation results seems incomplete to me. Do you see any other correlations? Why was 2015 different?
L318: What do you mean by “historical data”? Why do use a scaling factor of 0.7?
L328: If there are more appropriate values published for PAN deposition velocities (Wolfe, 2009), why do you use others?
L340: As already in L318, why do use this scaling factor?
L362-364: I am not convinced of this suggested correlation. For NOy, I think it is reasonable to state that there is some significant correlation with CCl4. However, for PAN it seems rather ambiguous whether the correlation is significantly stronger with ethane (R2=0.27) or with CCl4 (R2=0.35). Actually, I would say there is neither for ethane nor for CCl4 a good correlation to PAN.
L367: If there is no benefit from a multilinear regression, it is not necessary to discuss this. Please remove it also from table 2, since this only creating confusion.
L379: To me it does not look solely like scatter for the data below 2 ppb. In general, the ratio of i/n butane seems to be significantly larger for lower concentrations for any of the measured species. Is there any explanation for this?
L382-384: Where is the difference between PAN/NOy and the alkylnitrates? If I understand it correctly, all species exhibit i/n butane ratios similar to the DJ Basin.
Fig 7: In general, I would guess the strongest correlation here is observed for the i/n butane ratios and the wind direction. This should be pointed out more clearly in the discussion of the figure.
Fig 8: Units are missing in all panels!
L394: Units for CHBr are missing!
L390-394: I think you have to be careful here with the interpretation. As suggested above, it would be really beneficial to see some backward trajectories, since CHBr might also have other sources.

Additional comments

L195: I would suggest giving a wind direction instead of using the word “upslope”, since this seems a bit ambiguous. I would suggest changing this throughout the entire manuscript.
L200: What previous work? Please give a reference or show the data.
L282: I would recommend using SI units, i.e., kg N / m2
L288f: This is to general. Either you discuss possible future research directions or you better remove it.
L312: What is Vd?
L333: What other studies were publsihed?
L373 & Fig. 7: Is it Piceance or Pieance? The name in the figure differs from the one in the text.
L382 & Fig 7: I would suggest adding the letters of the panels to the text, where they are discussed.
L426-433: I would recommend removing the discussion on preliminary results from the conclusion, since you do not show any of these data.
L435: What previous work?

---

## Round 0.2 · accepted · Accept

I would like to thank you for your consideration of the comments made during the review process. This is an interesting and comprehensive data-set and makes a good contribution to the field. I was particularly interested in the balance in the deposition of reduced and oxidised RN components and the respective source regions.